# Challenges in Developing a Real-Time Bee-Counting Radar

**DOI:** 10.3390/s23115250

**Published:** 2023-06-01

**Authors:** Samuel M. Williams, Nawaf Aldabashi, Paul Cross, Cristiano Palego

**Affiliations:** 1School of Computer Science and Engineering, Bangor University, Bangor LL57 2DG, UK; 2School of Natural Sciences, Bangor University, Bangor LL57 2DG, UK

**Keywords:** *Apis mellifera*, honeybee, radar, machine learning, support vector machine, linear predictive coding, log area ratios

## Abstract

Detailed within is an attempt to implement a real-time radar signal classification system to monitor and count bee activity at the hive entry. There is interest in keeping records of the productivity of honeybees. Activity at the entrance can be a good measure of overall health and capacity, and a radar-based approach could be cheap, low power, and versatile, beyond other techniques. Fully automated systems would enable simultaneous, large-scale capturing of bee activity patterns from multiple hives, providing vital data for ecological research and business practice improvement. Data from a Doppler radar were gathered from managed beehives on a farm. Recordings were split into 0.4 s windows, and Log Area Ratios (LARs) were computed from the data. Support vector machine models were trained to recognize flight behavior from the LARs, using visual confirmation recorded by a camera. Spectrogram deep learning was also investigated using the same data. Once complete, this process would allow for removing the camera and accurately counting the events by radar-based machine learning alone. Challenging signals from more complex bee flights hindered progress. System accuracy of 70% was achieved, but clutter impacted the overall results requiring intelligent filtering to remove environmental effects from the data.

## 1. Introduction

Wild bees and honeybees both contribute more than USD 2900 ha^−1^ each to the production of insect-pollinated crops [1]. They are seen as critical for achieving sustainable development goals while being too poorly understood to capitalize on their potential [2]. The decline of managed honeybees and their keepers, as well as wild hives, has been documented [3,4]. Pressure is mounting to manage hives more effectively and with more consideration for their needs. Automating the counting of activity at the entrance to hives will provide detailed, live, and contextual information about their health and productivity.

Bee-counting devices capable of providing accurate data suitable for scientific inquiry are few. Most operate by using a type of camera to track bee traffic coming to and from the hive entrance [5]. Cameras can be either visual or infrared, and some studies have utilized capacitive sensors [6,7]. Radar has been used to monitor the signals reflected from bees and radar microphones to track bees through hive walls without disturbance [8,9]. However, fully automated, low-impact systems to achieve counting goals do not currently exist, with most systems requiring human input or modifications of the hive itself.

Previously published work undertaken by the authors suggests that radar systems can provide cheap, reliable, and simple-to-deploy bee counters [10,11]. This work differs in that it expands the problem to include background signal removal. In addition, the work uses multiple hives across different days to determine whether the system is resilient to the effects of weather change and clutter differences between hives.

Similar technologies have been used to monitor insect activity. Gaussian models have been used to address misreadings when counting bee behavior activity using RFID tags [12]. RFID has also been used with machine learning to determine insect species from activity at the entrance [13]. RFID is a powerful tool but relies on both tagging the bees and modifying the entrance of a beehive, limiting its use for wild and managed bees without disturbing behavior.

Zenith-pointing linear-polarized small-angle conical-scan (ZLC) entomological radars have been used to classify insect species based on weight, wing beat, and body length-to-width ratio [14]. X-band radar has been used alongside Support Vector Regressor algorithms to estimate insect mass based on each insect’s radio cross-section (RCS [15].) Radar has been demonstrated as a powerful tool for entomological purposes [8,16].

Machine learning using Doppler radar data captured from bees has not been otherwise investigated. Human activity has been classified using micro-Doppler signatures and machine learning [17]. Radar and machine learning have been investigated together for other animals, such as radar imagery being used to detect bird roosts using convolutional neural networks [18]. Lameness in farm animals has been automatically detected using machine learning classification of radar signatures [19]. The lack of research targeting bees using similar techniques leaves room for work tracking bee activity at the hive entrance using radar.

Bee movement tracking has been investigated using machine learning on data captured by a camera [5,10]. However, radar systems require less processing power, are cheaper, and can be more resilient to weather interference.

This study aimed to develop a real-time bee counting radar by integrating a Raspberry Pi © processor with a custom 5.8 GHz Doppler radar. This system fills a gap by allowing accurate counting of bee activity at the entrance of the hive. However, complex or overlapping bee flights created signals that could not readily be differentiated into the target classes. These challenges became the focal point of the study, providing a basis for continued development once these barriers are cleared.

## 2. Materials and Methods

### 2.1. Radar Receiver and Modelling Approach

The radar module supporting the present effort was similar to the 5.8 GHz continuous-wave (CW) radar Printed Circuit Board (PCB, JCLPCB, Hong Kong, China) deployed in [20] and is visible in Figure 1a. The PCB module integrated an in-phase/quadrature (IQ) mixer for the discrimination of positive and negative Doppler shifts. The IQ mixer fed 2 channels with identical 60 dB custom-designed Variable Gain Amplifiers (VGAs) and 100 dB common mode rejection ratio (CMRR) for amplification of the Intermediate Frequency (IF) signal. The VGAs additionally included a first-order low-pass filter limiting the IF output noise outside of the ~DC-408 Hz range. The VGA’s output was fed to a laptop using an external USB sound card with a 44.1 kHz sampling rate.

The received radar signals were well approximated through a simple model overlapping scattering from uniform speed translation of bee body and harmonic oscillation of an adjacent smaller scatterer, which mimicked wingbeat motion:(1)$$x_{r}\left(t\right)=A_{1}\cos\left(2\pi \frac{2}{\lambda }R\right)+A_{2}\cos\left\{2\pi \frac{2}{\lambda }\left[R+A_{H}\cos\left(\omega_{H}t\right)\right]\right\}$$

There, *A*_1_ and *A*_2_ represent the amplitude for the baseband body translation and wingbeat motion components, respectively, and are a measure of the respective radar cross sections (RCSs); *A*_1_ and *ω_H_* represent the wingbeat amplitude and angular frequency, respectively; *λ* represents the incident signal wavelength determined from the 5.8 GHz carrier; and *R* represents the radar-target range and was effectively independent of wingbeat motion in the extracted micro-doppler signatures scenarios where $A_{H}$ << $A_{2}$ << $A_{1}$. For typical experimental values of $A_{2}$ = $A_{1}$/5 = 0.2, *A_H_*~1 cm, *R* = 0.1–2 m, *ω_H_* = 2π(150–230) Hz, *λ*~5 cm, and bee speed ~0.2 to 2 m/s the body Doppler shift ranged between 2 and 20 Hz while sidebands from phase modulation in (1) were well visible up to the 400 Hz frequency range [20]. Conversely, setting $A_{1}$ = 0, and relaxing the $A_{H}$ << $A_{2\;}$ condition encodes an explicit dependence of range onto harmonic motion and enabled (1) to be used to model: the effect of radar shaking from wind (*ω_H_* ≤ 5 Hz); or mechanical coupling with a nearby (e.g., laptop fan) vibration source (*ω_H_* = 50 Hz). While (1) made higher frequency sidebands plausible, their prominence was expected to fade with increasing range because the VGAs output attenuates the IF signal components beyond 408 Hz.

A raw initial interpretation of the data was achieved by investigating the time-stamped radar signatures recorded of bees against a camera recording of transpiring events. Spectrogram representations of these data allowed for an initial assessment of the quality and detail recorded by the radar. Labels were provided for the events by a human observer.

These data were then processed by extracting features in the form of Log Area Ratios [21]. These features were the dataset used to train Support Vector Machine models to label new samples recorded by the radar [22]. The predicted labels were compared with those provided by the observer to provide an estimate of accuracy.

A final interpretation of results was achieved by comparing the accuracy of the generated models when predicting all labels for a separate, new recording against labels provided by the observer. This was to measure the effects of changes in environmental conditions on the ability of the model to predict correctly.

### 2.2. The Processing Equipment

The computing system was designed to minimize both cost and power consumption and was centered on a Raspberry Pi 4B. Without an AI Accelerator or equivalent, the Pi was not suitable for a deep learning approach. Instead, this system would leverage Support Vector Machines (SVMs [22]) to match previous work [11,20].

The sampling time was limited to 0.4 s. This window represents the smallest observed complete event in the original dataset. Even within 0.4 s, most recorded samples included one or more hovering bees as well as the other classes. In 4.4% of samples, both an inward and outward bee event took place within 0.4 s. This is true of overlapping inward and outward bees as well. A smaller percentage (0.18%) contained multiple overlaps such as two inward and one outward.

Other research studies, without machine learning or automatic counting, have placed the radar onto the hive surface, facing outward [8,23,24]. The approach was chosen to overcome the following challenges of such placements:It removed the need to modify the hive, which is advisable given that the system may be used on wild bees.Bees crawl at the entrance and may cover either antenna, as in Figure 2.Antennae have a radiation pattern that may cause flights to be lost from the detection cone if, for example, they walk to the edge of the hive before takeoff.While offering some protection against hovering bees, surface-mounted radar may still be obscured more infrequently.Limited research suggests that bees may be sensitive to the frequencies used and the equipment will function as a source of heat, which may affect behavior [25,26].

The position in this study ensured that the entire front surface of the hive was in view of the radar, and it was less invasive and the setup quicker. Hovering bees and weaker power reflection at the entrance of the hive remained an issue because of the free space between the radar and hive entrance.

Challenges were expected from the outset because there was no effort to standardize bee flights or control flight direction. Bees were free to leave in any direction, even crawling along the edge of the hive until takeoff on a side face. Similarly, on approach, bees could arrive from any angle and could be as quick or slow to enter as needed. When the entrance was congested, bees would often hover on arrival until there was space to enter, mimicking other hovering bees and obfuscating other activity when flying close to the antennae. The free-flying bees created complex radar samples that could not be intuitively labeled solely on signature structure alone.

Initial data were gathered across three days, consisting of twelve recordings with a maximum duration of 20 min each. Different hives were used during each day. Replacing the radar between sample gathering periods was not precise, because the system needed to be flexible, so long as it was placed within the expected range (1–2 m) of the hive as in Figure 1. When working with wild colonies it would not be possible to guarantee the same distance or angle, nor would it be advisable to force such placement if minimal disturbance was desired.

Each radar stream was accompanied by a video from a digital camera. The video recording was initiated first, and the radar data were aligned by the operator counting down to the commencement of the radar recording. This was suitable to align within half a second. Two or three clean bee events would, by matching video frames to timestamps, allow complete alignment.

This source dataset was gathered to train the algorithms. Once trained, these would then be used to label entire videos. The operator would provide corrections of the sample labels where needed and the resultant datasets fed into the training data pool.

The machine learning models would be expected to label entire videos. Therefore, an additional dataset was later included that featured one, full-length recording that was disaggregated into 0.4 s samples, and this was labeled and included in its entirety.

An overlapping window of 0.1 s was used to extract samples from consecutive or extended events, such as long hovering flights and background samples. A flexible approach was used when samples were not an ideal length for sub-division, modifying the final overlap to ensure all source data were used. For example, a signal of 0.6 s would be split into two 0.4 s samples with an overlap of 0.2 s.

Feature extraction for the primary system was achieved by using Log Area Ratios (LARs) derived from Linear Predictive Codes (LPCs) [21]. LPCs and their derivatives are a means of expressing the spectral envelope of a signal in compressed form. Their use in machine learning for radar data is relatively new and has successfully classified other, non-acoustic, signals [17,27,28].

The LARs were used to train a support vector machine with Bayesian hyperparameter optimization. Five different models were trained:Four-way classification.Background samples versus all others.Hover samples versus in and out.Three-way classification (hover, in, and out).Binary classification (in and out).

These models were chosen to allow multiple potential classification pathways. Either four-way brute classification, or splitting the problem into multiple, potentially easier, problems as demonstrated in Figure 3. These separate pathways were developed to maximize the opportunity for binary classifications that can favor SVM models [29,30].

To provide context, similar models to those in the authors’ previous work were used [11]. This was a DenseNet deep learning architecture with a custom head network [31]. This network would operate on spectrograms generated from the 0.4 s samples. While unlikely to be lightweight enough to run on portable hardware, this model would provide a crucial understanding regarding the suitability of the data for machine learning.

## 3. Results

### 3.1. Preliminary Results

Generated spectrograms of the signal samples provided evidence that signatures would have less information above 300 Hz (see Figure 4). This would exceed the typical flight speed of a bee at 8 m/s. As image processing networks require small inputs of no more than a few hundred pixels square, the authors limited the upper range of spectrograms to 300 Hz and then 150 Hz to maximize image quality. The change to 150 Hz was initiated as accelerating and decelerating bees were always much slower than their cruising speed and spectrograms contained little information above 150 Hz. Any information here was lost in the contrast limits of the generated images and would only penalize the models. Empty space in already small images would reduce the resolution of the lower-frequency, more powerful signatures.

However, results from the DenseNet deep learning approach were poor, with the best accuracy being 46.73%. Given the four-way nature of the problem, this is significantly better than a random choice, but the results warranted further investigation.

By using LARs, it was possible to achieve a preliminary accuracy of 75.12% in a four-way scenario (Figure 5). In all cases, a 9:1 split of training and testing data was used and the results were gathered as an average of tenfold cross-validation. The figure shows the outputs of running the experiment with three sets of data:**Set A:** the single channel, manually gathered Doppler data from the radar.**Set B:** the dual channel, manually gathered IQ data from the radar.**Set C:** the dual channel, complete IQ dataset including both the manual set and the full recording breakdown dataset.

Set C would be the dataset used in the testing phase of the work. This shows a performance penalty associated with fully captured datasets rather than hand-chosen samples. This is not unexpected, as more difficult samples (such as those with overlapping events) were required to be included. The results show that complete IQ datasets are more suited for machine learning than single-channel results.

Separating the problem into smaller challenges did not create better results. While background prediction is good (91.59%), this would then be followed by either hover prediction (83.19%) or three-way prediction (78.65%); together these would fall below base prediction accuracy (75.12%). The targets are the labels generated by the final classification, the inward and outward bees. Knowledge of background and hovering signals is useful but is not the goal of this work.

### 3.2. Exploring the Weaker Results

The weaker-than-expected results spurred a further investigation into the spectrograms generated. Complex signals, difficult to classify, became apparent due to the free-flying nature of the targets. Figure 6 shows an ideal sample of four consecutive outward flights of bees, which quickly accelerate toward the radar before passing by in proximity as confirmed by video recording. The first two flights overlap on the spectrogram, hindering the ability of the machine learning to count them separately as they exist in one 0.4 s window.

However, not all flights were clean. Figure 7 shows both a visual record and a spectrogram of complex overlapping events. These events are as follows:Takeoff for a single bee.Flight of the first bee to the right and behind the radar.A hovering bee emerges from under the radar and flies off-screen to the left.Vertical takeoff of two bees, one does not approach the radar.The second of the two bees loops, increasing speed, and exits the frame.The inward bee from the screenshot appears.The first of the three bees in the screenshot takes off.Two more bees take off after the first.Closest approach of the exiting bees.Inward bee enters the hive.The last view of the exiting bees, flying away from the radar both left and right.

While the signal happened across eight seconds and would be broken down into smaller, easier-to-classify samples, there is a paucity of information when multiple overlapping events took place. Specifically, between events 7 and 10, there is a compounding of the signals, justifying that the spectrogram approach would be met with failure.

Some events were too similar in the target frequencies to separate visually. An example of these is provided in Figure 8. The first event (a) is of a hovering bee that moves both towards and away from the radar with variable speed. The second event (b) is two inward bees flying towards the entrance of the hive; however, there is a sudden uplift of wind, which makes their flight difficult, and they struggle to fly along a fixed path.

Figure 9 shows three hovering bees, none of which enter the hive or leaves the area during the segment. At 0.4, 0.75, and 1.5 s some examples are like the outward signals present in the ideal sample. Multiple hovering bees in a signal recording were common.

These signals are a close visual match to other, less ideal outward signals. In the samples collected, there were matches between all four classes. A spectrogram deep learning approach would encounter a point of no improvement due to the restraints of the visualization format. In the future, as this dataset is expanded, the visual overlap will continue to grow.

Given this limitation, questions emerged regarding the signal compression techniques and mild success. To understand how the data allowed the models to perform well, several exploratory investigations were undertaken.

The major disparity between these results and others found in literature was the number of LARs used in this work. It is common to expect 10 or fewer LP coefficients (equivalent in number to LARs) for each small window, itself less than 100 milliseconds [17].

In contrast, the models required that the 400-millisecond signal not be subdivided, and as such, the number of coefficients climbed at first to 240 per window for a 44.1 KHz sample rate and 100 per window for a down-sampled 3.5 KHz rate. This high number of coefficients is problematic. As the number of coefficients increases the algorithm quickly includes noise from the source.

Using the full number of coefficients, accuracy response as a function of the sample rate was assessed. The results are presented in Figure 10. This shows that accuracy required a sampling rate of greater than 3 KHz to achieve a plateau of growth. The exception to this was predicting background and binary signals, which had a strong response from any sampling rate, expected as these are simpler predictions.

To investigate signal sub-division to match other works in the literature, the Raspberry Pi © was first benchmarked to confirm limits to the number of coefficients that could be used. The results are presented in Table 1 and ‘times required’ have been measured to include running a prediction. This is to ensure the process happens faster than the 0.4 s window.

Generating many coefficients for a 0.4 s window is computationally taxing. By using multi-core processing to handle each channel separately, the Raspberry Pi could encode 240 LARs in a 0.35 s window.

With these limits, a benchmarking routine was created to determine accuracy as a measure of the sub-window size and number of coefficients. The experiment was also conducted when downsampling the signal to 3 KHz and 1 KHz to measure whether lower frequency components become more important when sub-dividing the window.

The findings are presented in Figure 11, demonstrating that the sub-division of the sample window decreases accuracy. For completeness, all sub-window lengths with the full 240 LARs are included, which would not be possible to run in real time on the Raspberry Pi. Even with all coefficients available, LPC derivative machine learning accuracy decreases as the signal is segmented. As LPCs are compression techniques, it can be understood that segmenting the signal further decreased the information in each resulting window. A comparison would be the segmentation of four similar spoken words into small time windows, which would decrease the overall context included as opposed to encoding the entire words with one compression window.

It became clear that there were either high-frequency and/or low-power components to the signals that were not easily shown on a spectrogram. These elements were crucial for machine learning success. The signal could not be further segmented without decreasing accuracy. Together, these findings supported that these components are being obscured by background noise.

It had been an expected evolution of the work to begin creating filtering algorithms to strip out the clutter associated with outdoor recordings in variable weather. However, the complexity of the filters will now become more challenging. Preserving complex patterns while removing the effects of wind and other clutter will be challenging.

However, without filtration, the machine learning models would be unlikely to adapt to new recordings. The existing data were recorded as subsets each from a single or group of videos, each with its own setups and environmental conditions. This could be introducing noise into the dataset, which meant that models were unprepared for new sets of data from previously unseen conditions.

The following question was whether leaving the sampling rate at the maximum 44.1 KHz was introducing needless noise that was affecting the feature encoding stage. Another routine was designed to measure how accuracy reflected the number of coefficients at differing sample frequencies. Lowering the sampling rate decreases accuracy, as shown in Figure 12. However, at lower sampling frequencies accuracy requires fewer encoding coefficients. A notable plateau is present at 100 coefficients or more with a sampling frequency of 3.5 KHz, followed similarly by other sampling frequencies with the same number of coefficients.

While these results compare poorly to allowing an unrestricted sampling frequency, they show that the models require fewer LARs at lower frequencies to achieve maximum accuracy. This could indicate that the models may have been learning more general patterns in the data when given lower sampling frequencies to work with. When running the final tests, the results of lower-frequency, fewer-coefficient encoding would be included to measure whether models could become more generalized.

Now that it had been determined that the models were not influenced by noise included with an unrestricted sampling rate, it became prudent to analyze the signals in greater depth. LPCs are a compressed form of the spectral envelope of a signal. As such, it was useful to generate the spectral envelope for each signal and produce a standard deviation per class. In Figure 13, the standard deviation of all spectral envelopes in each class is shown up to 1.5 KHz. Standard deviation is shown, as averaging spectral envelopes would remove most peaks.

The standard deviation in the background class is the flattest, except for several peaks centered at 1 KHz, which is faint noise in the signals, often masked by the bees themselves, caused by the recording equipment.

The bees themselves are visible as a strong peak of deviation at sub 150 Hz frequencies, matching the signatures seen on spectrograms. Outward signals have a peak slightly higher in frequency, which can be explained by bees rapidly accelerating away from the hive. Inward bees decelerate and hovering bees are unlikely to reach a maximum speed near the hive. Notable peaks can be seen at 400 Hz and 800 Hz. Smaller peaks can be seen throughout, some more pronounced in one class over others but these are minor.

### 3.3. Testing Stage

The machine learning was assessed on its accuracy in predicting the entire test set with all other data included as learning data (Figure 14). Significant penalties when using a separate setup are apparent. When exposed to new data, from a new radar position in differing conditions, the models lose their capabilities. Four-way classification accuracy drops to 70%, with a precision of 0.63 and recall of 0.70 due to imbalanced class sizes.

Sets in this figure are as follows:**Set A:** the complete training dataset was used, sampled at 44.1 KHz with 240 LARs.**Set B:** the complete training dataset was used, sampled at 3.5 KHz with 100 LARs.**Set C:** the smaller, manually extracted dataset with higher training accuracy was used, sampled at 44.1 KHz with 240 LARs.**Set D:** the smaller, manually extracted dataset with higher training accuracy was used, sampled at 3.5 KHz with 100 LARs.

For completeness, the results for a down-sampled dataset at 3.5 KHz with 100 coefficients are included. Overall accuracy improved by 1–12% despite the lower training accuracy. A critical note for the four-way classification is that no inward bees were predicted correctly (121 samples or 4.8% of the data to label.) The figures for this four-way classification are skewed by the much larger hover and background classes. This is evident when looking at the F1 macro scores, which expose accuracy bias caused by imbalanced classes.

Set B outperformed Set A despite lower training-stage results. This supports that different frequency bands and coefficient numbers benefit some classifications despite lower training accuracy. While adding more recordings, from differing weather and hive conditions, will improve the results further, the results above suggest that future gains will be ever-diminishing.

To achieve complete capability in this system, filters are a requirement. These filters will be challenging because of the complex signatures that form part of the machine-learning process.

## 4. Discussion

Compared to previous work by the authors, the results from this work are poorer [20]. For three-way classification, 93.37% accuracy was achieved, and 91.13% binary accuracy was achieved in the last effort. Similar results for this work were 81.67% and 88.33% accuracy for three-way and binary classification respectively (see Figure 5).

However, some key changes in the experimental setup explain the differences. This study used no data augmentation as the volume of data was considered sufficient. Data augmentation improves smaller datasets by creating a larger pool for training but can also make a set more homogenous and therefore easier to classify. The data recorded here were gathered across multiple days from more than one hive, which differs from previous studies where one hive was used on one day. The changes in radar distance and angle, coupled with varying weather, introduce more difficulty. These additional challenges were inevitable in the development of a real-time implementation radar classification system.

Nevertheless, the expected outcome of this study was to meet or exceed previous results. Without this being achieved, there is further work remaining to overcome the shortcomings highlighted in this study.

The closest study in the literature to this work comes from Souza Cunha et al. in 2020 [8]. This study used the root mean square (RMS) of a Doppler radar as a measure of activity at the hive entrance, validating this by manually counting bees during recordings using a handheld clicker. RMS has key benefits as it is a simple, non-ML approach that gives a good measure of activity, which they were able to show correlates to hive health. As such, this approach is closer to field deployment readiness than the work here. However, they admit that ‘non-foraging’ bees (equivalent to hovering bees in this work) are counted in the RMS signal and there is no discernment between inward and outward bees using the radar. Our work is an attempt to overcome these limitations and once fully developed will provide more precise information for future study.

The results show a pattern in that so long as sufficient data are available for each hive, distance, and weather condition, then the models are reasonably accurate. As soon as new conditions are introduced, the models lose accuracy. This is not unexpected, but the degree to which minor signal elements are necessary for good classification was not anticipated. These minor elements would too easily be removed by simple filters for environmental conditions.

Hovering bees introduce unique challenges in that, given the resolution of the radar, they appear to mimic the flights of other bees. They do this by passing close to the entrance of the hive while accelerating or decelerating, but not stopping. Minor differences in the signals will be useful to detect the difference between a slowing bee and one that stops. Again, these differences will be subject to interference from the environment.

Despite lower performance during initial training, models trained on subsampled signals with fewer LARs performed better than those with the complete data. This supports the interpretation that the bulk of useful information is contained at lower frequencies. This is also shown when investigating the spectral envelope of each class, which shows more deviation at lower frequencies. However, the identification of which exact frequency bands are most important is challenging. Further work could look at performing statistical analysis of the signals in depth. This could provide guidance when developing filters as to which frequency bands are most important.

Hand-picked samples provided better training accuracy than the dataset containing all available data. The dataset containing all the data was more useful at the test stage. This is evidence that a hybrid approach may be useful in the future, with a dataset containing a core set of hand-chosen, clearer samples to provide a strong foundation. This is in addition to containing entire recording breakdowns, which will provide many hard-to-classify ambiguous samples.

This work is useful as no similar attempt has been made to classify honeybee activity at the entrance of a beehive using Doppler radar. Early experiments such as the one presented are necessary to identify the limits of existing technologies and algorithms as well as provide guidance for overcoming such restrictions.

This research implies that further work is needed to create a deployable real-time radar. A greater understanding of radar bee signatures is required so that good filtration can be enacted that does not remove the weaker signal elements.

## 5. Conclusions

An investigation into generating machine learning models to classify real-time radar data on honeybees has been detailed. These models aimed to monitor and count activity at the entrance to the beehives. Data gathered in this fashion, which are automatically labeled by machine learning models, would provide valuable data for ecological research and for businesses looking to improve their use of honeybees. The models generated in this work achieved an accuracy of 70% though, by other metrics, the class imbalance created biased results.

Data were gathered from multiple hives across a few days from beehives kept at a farm. The data were split into 0.4 s samples, labeled by using video camera recordings of each event, and transformed into Log Area Ratios. These were then used to train Support Vector Machines to predict labels for new samples.

Challenges in progressing further have been identified. It is argued that a filter is needed, as high-frequency, weak signal elements appear to be needed for successful classification. These high frequencies are subject to interference and contain weak signal components that will be difficult to preserve. A greater understanding of these weak signal components is needed.

The limits of this work are clear. Four days of data were used from a small selection of beehives. To develop the solution further, many more hives would be required. Data would need to be captured that reflected all feasible weather conditions. Some, such as rain, may render the system incapable of predictions at all. In addition, an intelligent filter must be investigated to provide a means of removing much of the radar clutter that is unavoidable when recording outdoors while preserving weak but vital signal elements.

No further machine learning work is advised until filters are developed. Though additional data will result in increased accuracy, the system will not be resilient until environmental changes can be addressed. This work has functioned to provide specifications that future filters will need. With suitable further study, the work supports that the capability will exist to classify honeybee activity in real time.

## Figures and Tables

**Figure 1 sensors-23-05250-f001:**
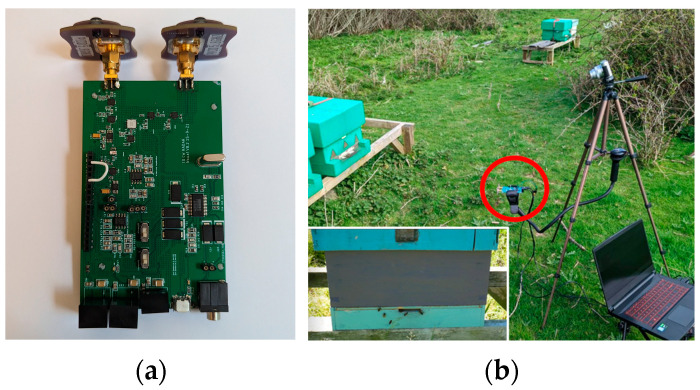
(**a**) The radar used during this experiment and (**b**) Experimental setup (radar encircled in red) and an example of a standard hive and nuc (nucleus colony) box.

**Figure 2 sensors-23-05250-f002:**
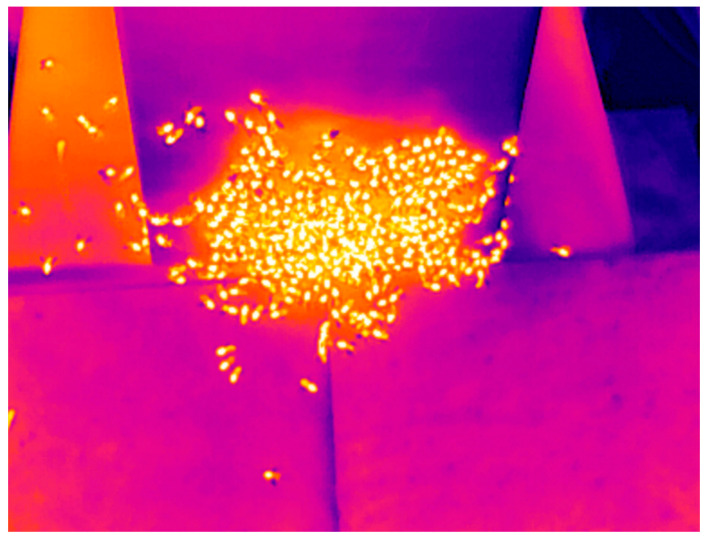
A thermal imaging camera capture of bees crawling over the entrance of a busy hive.

**Figure 3 sensors-23-05250-f003:**
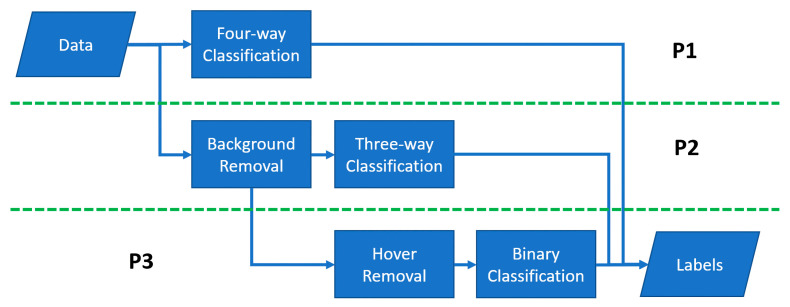
Three prediction pathways (P1, P2, P3) toward labeling samples. Continued binary classifications may favor Support Vector Machine (SVM) architecture over multi-class problems.

**Figure 4 sensors-23-05250-f004:**
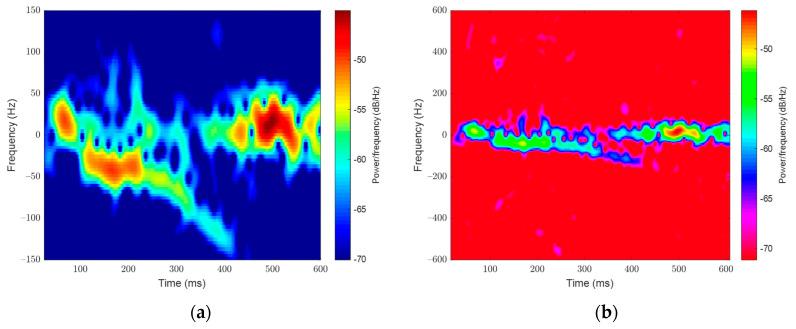
(**a**) A complete signal sample (outward bee) spectrogram limited to 150 Hz matching the images that were inputted into the deep learning models. (**b**) A larger range, high contrast spectrogram of the same signal shows a paucity of information beyond 150 Hz.

**Figure 5 sensors-23-05250-f005:**
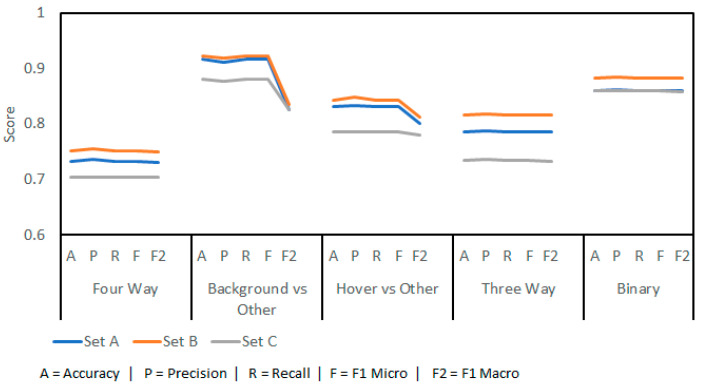
Results from preliminary machine learning models using Log Area Ratio (LAR) implementation.

**Figure 6 sensors-23-05250-f006:**
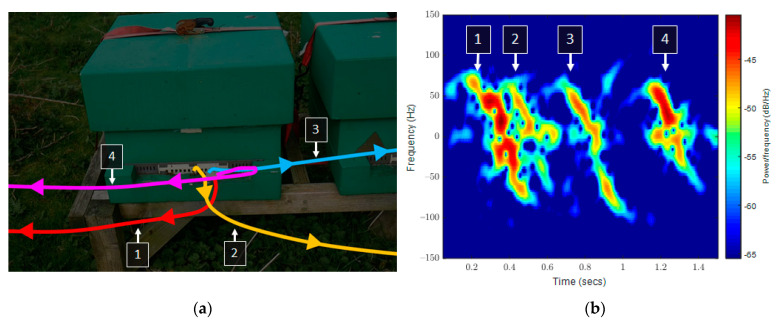
(**a**) Image showing the trajectories of four bees and (**b**) spectrogram recording of this event. The first two overlapped, limiting attempts to separate them. The numbered lines are the four flights recorded both in the video and spectrogram.

**Figure 7 sensors-23-05250-f007:**
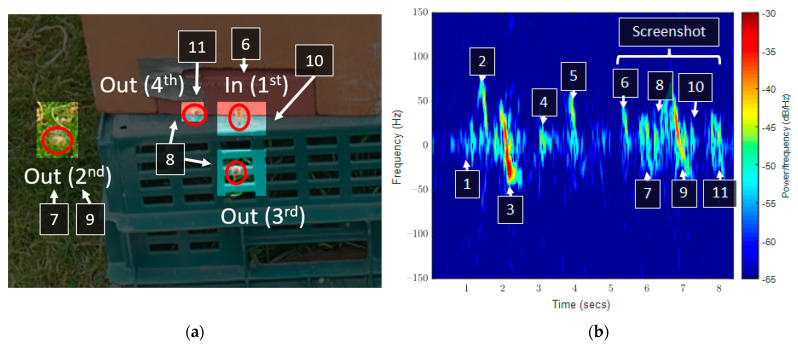
(**a**) A screenshot of the video recording of an event and (**b**) the corresponding spectrogram representation of the signal, showing complex overlapping elements. The red circles are the bees recorded across the events and the numbered items show correlation between spectrogram and the video recording.

**Figure 8 sensors-23-05250-f008:**
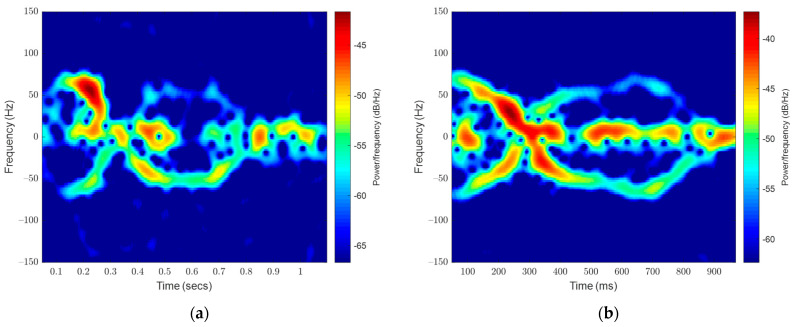
Two signals (**a**) showing a hovering bee signal and (**b**) showing an inward bee signal.

**Figure 9 sensors-23-05250-f009:**
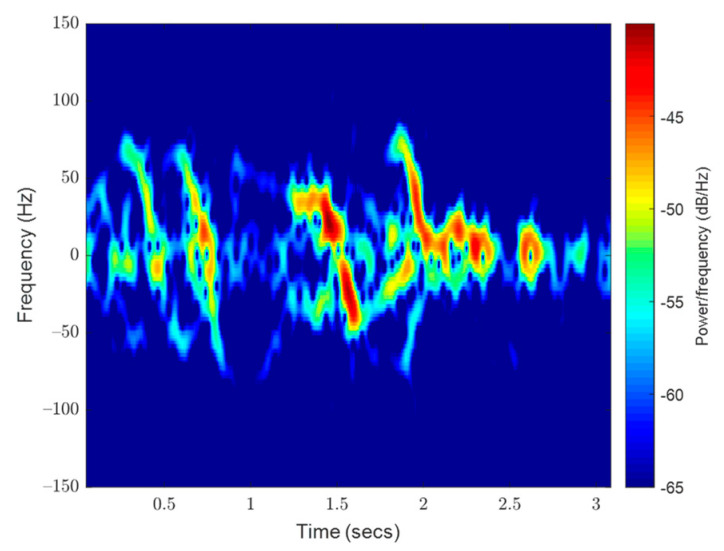
A hovering signal of three bees shows similarities to outward bee signals.

**Figure 10 sensors-23-05250-f010:**
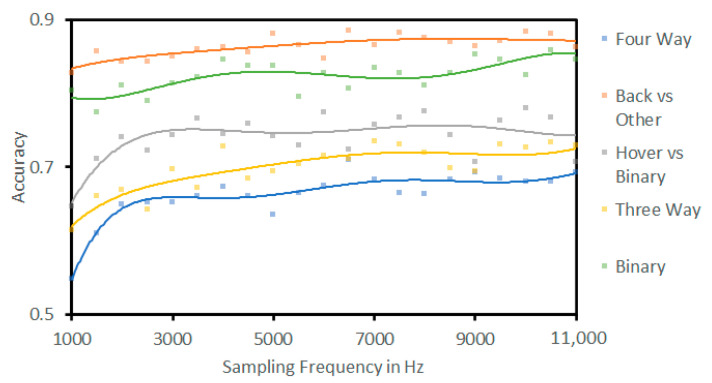
Accuracy versus sampling rate across the different prediction pathways, showing that accuracy changes in response to varying the sampling rate of the signal.

**Figure 11 sensors-23-05250-f011:**
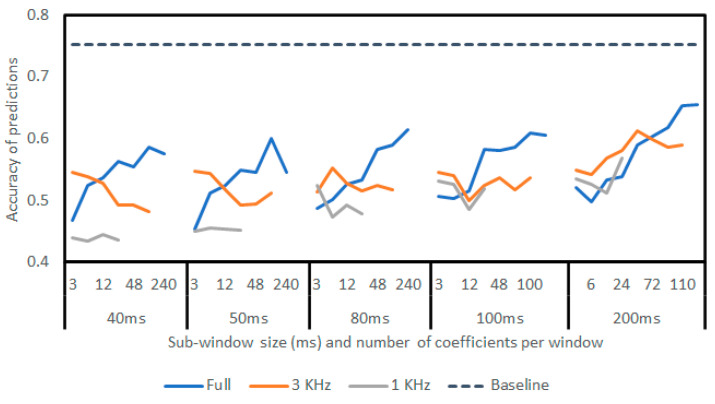
Results from sub-windowing the signal with differing coefficient numbers. Includes accuracy at 44.1 KHz sampling rate and change in accuracy at both 3 KHz and 1 KHz. At 1000 Hz, some window/coefficient combinations could not be run due to insufficient data.

**Figure 12 sensors-23-05250-f012:**
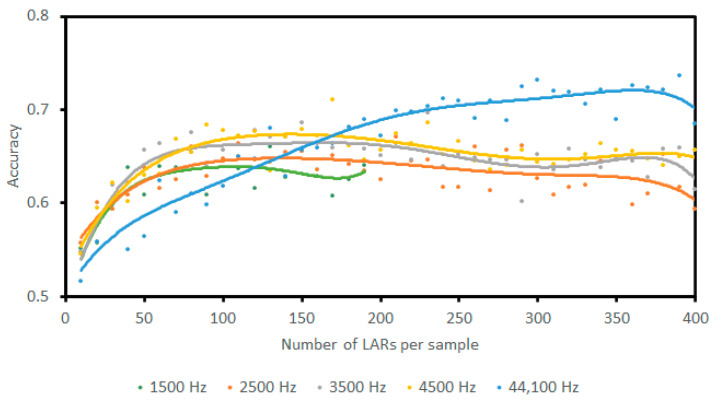
Accuracy versus the number of encoding coefficients for a range of sampling frequencies. Legend indicates sampling frequency in Hz. When using a 1.5 KHz sampling rate, it was not feasible to include large numbers of coefficients as the data became sparse.

**Figure 13 sensors-23-05250-f013:**
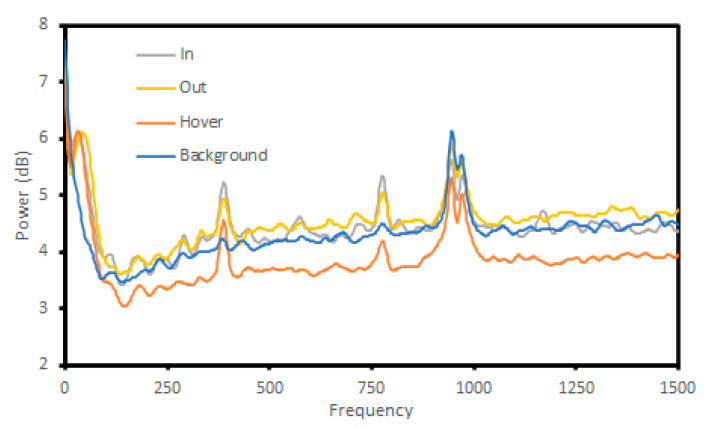
The standard deviation of the spectral envelopes for each class.

**Figure 14 sensors-23-05250-f014:**
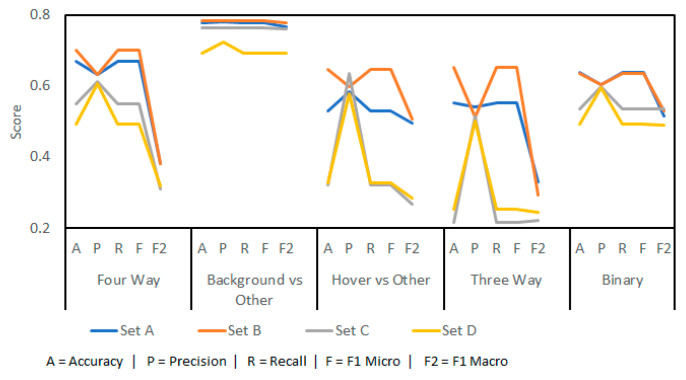
Testing results from the final stage that show a decrease in performance versus the preliminary results. This is an effect of recording in outdoor spaces with variable conditions.

**Table 1 sensors-23-05250-t001:** Possible sub-window sizes on the Raspberry Pi © and the maximum number of coefficients per window possible.

Sub-Window Size	Encoding Limit	Total Number of Features per Channel	Time Required
40 ms	76	760	350 ms
50 ms	84	672	348 ms
80 ms	96	480	349 ms
200 ms	110	220	352 ms
400 ms (full window)	240	240	351 ms

## Data Availability

Data supporting the reported results is collected in the OpenAIRE Repository: https://zenodo.org/communities/specialinsight/ (accessed on 31 May 2023).

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
