# Peer review of "Challenges in Developing a Real-Time Bee-Counting Radar"

_sensors, 2023, doi:10.3390/s23115250_

Round 1
Reviewer 1 Report
The manuscript is well written and easy to read taking into account that he information is difficult because, they mex radar detection in apicultural practice. I believe this manuscript is relevant because provide novel information related to real-time radar data on honeybees.
Author Response
The manuscript is well written and easy to read taking into account that he information is difficult because, they mex radar detection in apicultural practice. I believe this manuscript is relevant because provide novel information related to real-time radar data on honeybees.
The authors are grateful for this feedback and are encouraged to improve the manuscript.

Reviewer 2 Report
1. There are some grammatical, alignment and typographical errors noted in the manuscript and it should be thoroughly checked and corrected throughout the manuscript. For example,
· in line number 32, the words “form of” may be as “form”;
· in line number 54, “first order” as “first-order”;
· in line number 364, “maximum” as “a maximum”;
· in line number 371, “set up” as “set-up;
· in line number 404, “beehives” as “the beehives”.
2. When writing an abstract please keep it short but cover all the contents of the manuscript. The authors have given the purpose of this research. But, there is no background why this research was conducted has not been implied in the manuscript is not given clearly and it should be rewritten.
3. The authors should improve the quality of the images (Figure 3) used in the manuscript with high resolution for better understanding. For example, the letters used are blurred and it should be rectified.
4. The discussion section is not given and no emphasis is given on the discussion of the results like why certain effects are coming into existence and what could be the possible reason behind them. After the results section the authors have given only conclusion.
5. When writing discussion, the authors may discuss of their results by focusing on the present findings and introducing data from other authors who also worked with the same or other studies with recent references.
6. The conclusion seems very simple. All conclusions must be convincing statements on what was found to be novel, impact based on the strong support of the data/results/discussion. Moreover, the authors may also be included the limitation of the present findings for a better understanding of the manuscript.
Author Response
- There are some grammatical, alignment and typographical errors noted in the manuscript and it should be thoroughly checked and corrected throughout the manuscript. For example,
- in line number 32, the words “form of” may be as “form”;
- in line number 54, “first order” as “first-order”;
- in line number 364, “maximum” as “a maximum”;
- in line number 371, “set up” as “set-up;
- in line number 404, “beehives” as “the beehives”.
Thank you. We corrected the issues with the grammar and language, line by line as given.
- When writing an abstract please keep it short but cover all the contents of the manuscript. The authors have given the purpose of this research. But, there is no background why this research was conducted has not been implied in the manuscript is not given clearly and it should be rewritten.
We are keen to improve our abstract to your expectation. The abstract was rewritten with a focus on methods and motivations.
- The authors should improve the quality of the images (Figure 3) used in the manuscript with high resolution for better understanding. For example, the letters used are blurred and it should be rectified.
We have replaced Figure 3 with a higher resolution copy.
- The discussion section is not given and no emphasis is given on the discussion of the results like why certain effects are coming into existence and what could be the possible reason behind them. After the results section the authors have given only conclusion.
We agree an isolated discussion section is required. We have added a discussion section to match expectations, this breaks down our interpretation of the results. Please see lines 445-493.
- When writing discussion, the authors may discuss of their results by focusing on the present findings and introducing data from other authors who also worked with the same or other studies with recent references.
We agree that we need to create a stronger comparison to both our previous work and other authors’. There are additional radar studies now added to the expanded literature review (lines 43-69) which used radar to monitor bee behaviour. However, these do not use machine learning to automatically classify activity. In addition, similar studies have used higher frequency radar than the more flexible 5.8 GHz system we used. While higher frequency radar does provide higher resolution recording of events, which in turn could improve machine learning capabilities, there are a few limitations. Signal attenuation is much more severe which would diminish range significantly and cost increases drastically at higher frequencies. Both would make any automated system less flexible than our chosen approach. We do have previously published work (references 11 and 19) which were early investigations into this problem. We have added a discussion as per the previous request. In this discussion we directly compare the results of this study with our previous work, highlighting the lesser performance while factoring in the more challenging circumstances.
- The conclusion seems very simple. All conclusions must be convincing statements on what was found to be novel, impact based on the strong support of the data/results/discussion. Moreover, the authors may also be included the limitation of the present findings for a better understanding of the manuscript.
We can see the reviewer’s point of view and agree that the conclusion needs improvement. We have largely rewritten the conclusion to better discuss limitations and highlight key findings. Please see lines 497— 516.

Reviewer 3 Report
Present manuscript "Challenges in Developing a Real-time Bee-counting Radar" is attractive and applicable. it is well structured but the Introduction is very weak and a comprehensive description of the research literature is not given. You should first point out a sufficient number of related studies and then point out the distinct difference of your research.
It is even recommended that at the end of the results and discussion section, the quantitative difference between the results obtained in this research and previous researches should be mentioned in a tabular format.
The English writing of the article was generally good and all the content was understandable for the reader
Author Response
Present manuscript "Challenges in Developing a Real-time Bee-counting Radar" is attractive and applicable. it is well structured but the Introduction is very weak and a comprehensive description of the research literature is not given. You should first point out a sufficient number of related studies and then point out the distinct difference of your research.
We feel that this response is in tandem with the editor. As such, we have added a large section (see lines 43-69) to the paper detailing the literature around machine learning, radar, and insects. Radar has been used to predict insect species, and RFID has been used to monitor activity at the beehive entrance. This paper (and its immediate predecessors by us) are the first documented case of using Doppler radar for machine learning-based activity counting at beehive entrances.
It is even recommended that at the end of the results and discussion section, the quantitative difference between the results obtained in this research and previous researches should be mentioned in a tabular format.
We agree that a direct comparison between the current work and our previous research is needed. We have added a discussion section (lines 445-493) which includes direct comparison. We believe that context is important as the previous studies used different setups in a less complicated environment. They did achieve better results when comparing explicit results (a peak accuracy for direct experimental comparison being 93.37% versus 81.67%) from one machine model to another, however, in the context of the overall capability to predict bee presence, counting, and behavioral interpretation no model performed optimally across all possibilities. The models in this paper were trained on a significantly harder dataset that aimed to capture more of the challenges of predicting bee behavior across multiple days and weather conditions. As such we have kept this discussion in text rather than tabular to provide this contextual comparison.

Round 2
Reviewer 2 Report
1. There are some grammatical, alignment and typographical errors are noted in the manuscript and it should be thoroughly checked and corrected throughout the manuscript. For example,
· in line number 122, the words “changes of” may be as “changes in”;
· in line number 339, “down sampling” as “downsampling”;
· in line number 123, “ability for” as “ability of”.
2. The figure legends should be improved and a proper footnote should be given. All legends should have enough description for a reader to understand the figures without having to refer back to the main text of the manuscript. For example, the necessary abbreviations should be given (For example, SVM).
3. The authors may improve the discussion of their results by focusing on the present findings and introducing data from other authors who also worked with the same or other studies with recent references since it is lack of sufficient references.
Author Response
We thank the reviewer for the kind support and contribution in improving the manuscript.
Reviewer 3 Report
The satisfactory revision was made on content.
Author Response
Comments and Suggestions for Authors
- There are some grammatical, alignment and typographical errors are noted in the manuscript and it should be thoroughly checked and corrected throughout the manuscript. For example,
- in line number 122, the words “changes of” may be as “changes in”;
- in line number 339, “down sampling” as “downsampling”;
- in line number 123, “ability for” as “ability of”.
We have revisited the manuscript and carefully searched for all alignment, typographical and grammatical errors. We have corrected these (highlighted in red) in addition to those suggested directly.
- The figure legends should be improved and a proper footnote should be given. All legends should have enough description for a reader to understand the figures without having to refer back to the main text of the manuscript. For example, the necessary abbreviations should be given (For example, SVM).
We have adjusted several figure captions to be more self-contained, explaining any abbreviations so that they are understandable without the main text.
- The authors may improve the discussion of their results by focusing on the present findings and introducing data from other authors who also worked with the same or other studies with recent references since it is lack of sufficient references.
We have taken the most comparable, non-machine-learning work available (Souza Cunha et al. 2020) and expanded on the discussion text by talking about the differences between our work and this study. While they don’t use machine learning, making direct comparison impossible, their approach is novel and capable. Please see lines 463-472. If there are any specific items of literature it is felt would improve our discussion further, we are happy to receive a reference and investigate a comparison.
